# Large Language Models: The Need for Nuance in Current Debates and a Pragmatic Perspective on Understanding

**Bram van Dijk[1], Tom Kouwenhoven[1], Marco Spruit[1,2],** and **Max van Duijn[1]**

[1]Leiden Institute of Advanced Computer Science
[2]Leiden University Medical Centre
{b.m.a.van.dijk, t.kouwenhoven, m.r.spruit, m.j.van.duijn}
@liacs.leidenuniv.nl

## Abstract

Current Large Language Models (LLMs) are unparalleled in their ability to generate grammatically correct, fluent text. LLMs are appearing rapidly, and debates on LLM capacities have taken off, but reflection is lagging behind. Thus, in this position paper, we first zoom in on the debate and critically assess three points recurring in critiques of LLM capacities: i) that LLMs only parrot statistical patterns in the training data; ii) that LLMs master formal but not functional language competence; and iii) that language learning in LLMs cannot inform human language learning. Drawing on empirical and theoretical arguments, we show that these points need more nuance. Second, we outline a pragmatic perspective on the issue of 'real' understanding and intentionality in LLMs. Understanding and intentionality pertain to unobservable mental states we *attribute* to other humans because they have *pragmatic value*: they allow us to abstract away from complex underlying mechanics and predict behaviour effectively. We reflect on the circumstances under which it would make sense for humans to similarly attribute mental states to LLMs, thereby outlining a pragmatic philosophical context for LLMs as an increasingly prominent technology in society.

## 1 Introduction

The performance of Large Language Models (LLMs) has recently reached high levels (see e.g. Bommasani et al., 2021; Mahowald et al., 2023). LLMs are deep neural networks with a Transformer architecture (Vaswani et al., 2017), trained to predict masked words from context, using massive text datasets.[1] During training, LLMs learn to represent

input syntactically in hierarchical form, and they also learn semantic relations (Rogers et al., 2021), which are useful features in summarising, question-answering, and translating text. Examples of recent LLMs are LLaMA (Touvron et al., 2023), GPT-3 & 4 (Brown et al., 2020; OpenAI, 2023), and PaLM (Chowdhery et al., 2022).

LLMs have sparked a lot of debate, inside and outside academia, around the question what their successes and failures say about linguistic capacities in AI systems, but also in humans. In the first part of this paper, we scrutinize three key points recurring in arguments by experts critical of LLM capacities (e.g. Bender and Koller, 2020; Bender et al., 2021; Browning and LeCun, 2022; Marcus, 2022; Shanahan, 2022; Mitchell and Krakauer, 2022; Bisk et al., 2020; Floridi, 2023; Mahowald et al., 2023), although there are also experts who are more optimistic on this matter (e.g. Sahlgren and Carlsson, 2021; Agüera y Arcas, 2022a,b; Berger and Packard, 2022; Cerullo, 2022; Sejnowski, 2022; Piantadosi and Hill, 2022; Piantadosi, 2023). These points are:

1. that all LLMs can do is predict next words;
2. that LLMs can only master formal as opposed to functional language competence;
3. that language learning in LLMs cannot inform human language learning.

In the first part of this paper, we aim to nuance these points and show that they are hard to maintain in the face of empirical work on LLMs and theoretical arguments. In the second part, this leads us to develop a pragmatist perspective on LLMs, for which we draw on work by Daniel Dennett, Richard Rorty, and others. 'Real' language understanding and intentionality consist of *attributions* of unobservable mental states, that humans make on the basis of observable behaviour. We do so because this has *pragmatic value*: it simplifies complex underlying biophysical processes and allows us to predict future behaviour. Instead of asking whether LLMs

---

[1]Some LLMs additionally benefit from further fine-tuning, e.g. reinforcement learning from human feedback (Christiano et al., 2017). Since evidence is emerging that most of LLMs' capabilities are learned during pre-training (Gudibande et al., 2023; Ye et al., 2023; Zhou et al., 2023a), we abstract away from this aspect in this paper.

have 'real' understanding and intentionality, we ask under what circumstances regarding LLM behaviour and their role in society, it is reasonable for humans to make mental models of LLMs that include capacities like understanding and intentionality.

In sum, our aim is to contribute to a more realistic framework for understanding LLMs within academia and beyond, which is better grounded in empirical and philosophical work. Since LLMs' impact on research and society likely increases in the future, properly understanding them is key. Still, although we defuse various critiques of LLMs in this paper, it is not our purpose to advocate their deployment without ongoing reflection on their implications. Examples include their environmental impact (Bender et al., 2021), biases (Lucy and Bamman, 2021), problems for educators (Sparrow, 2022), and ethical issues in adding human feedback (Perrigo, 2023), but these issues go beyond the scope of this paper.

## 2 Key points in debates on LLMs

In this section, we qualify three key points from the debate regarding LLM capacities by drawing on theoretical and empirical work.

### 2.1 All LLMs can do is next word prediction

Shanahan (2022) claims that whenever we prompt a LLM, for example with 'The capital city of the Netherlands is ...', we *actually* ask 'Given the statistical patterns you learned from dataset Y during training, what word is most likely to follow the provided sequence?', where the answer is likely 'Amsterdam'. According to this assumption, this is all there is to it; we should not speak about the model's potential topographical knowledge, nor should we say that the model understands the question in any way comparable to how humans understand it. We can see similar claims in Bender et al. (2021); Marcus (2022); Chomsky (2023); Floridi (2023), and in weaker form in Mitchell and Krakauer (2022).

Although it is true that the good performance of LLMs on many tasks stems from a simple training objective, which is predicting masked words from context, we argue that this point overlooks the complex ways in which LLMs are able to represent information. During training, LLMs induce various semantic and syntactic features that the model uses internally to *represent* the input in a manner that can be extracted, for example, by analysing

model weights or patterns of neuronal activation. An example regarding syntax is that LLMs are able to hierarchically represent input (Hewitt and Manning, 2019; Manning et al., 2020; Rogers et al., 2021; Mahowald et al., 2023). That is, they are capable of internally parsing the example into syntactic chunks, such that the prepositional phrase ('of The Netherlands') provides information about the noun phrase ('The capital city'), which constitutes a clue to the answer ('Amsterdam'). In addition, regarding semantics, the vector representations of words that neural networks induce are shown to be context-sensitive and rich enough to capture conceptual relations in line with human judgements (Reif et al., 2019; Grand et al., 2022; Piantadosi and Hill, 2022). Moreover, in our example, 'The Netherlands' and 'Amsterdam' are likely geometrically related in the vector space of a LLM, which provides a further clue regarding the answer.

Our syntactic and semantic examples here are not necessarily the way LLMs represent the relevant linguistic information; it is not trivial to extract representations from LLMs (Rogers et al., 2021). The point is rather that LLMs are capable of further *representing* input in various ways that are *not reducible* to either their training data or objective (Piantadosi and Hill, 2022). While they are not explicitly trained to represent input hierarchically, or represent semantic relations, such properties emerge while becoming better at their relatively simple stochastic training objective (Manning et al., 2020). On second thought, this should not be too surprising, given that there is a lot of linguistic information 'hidden' in the web text used to train LLMs, which LLMs (partially) reconstruct. Note that this argument does not require LLMs to 'really' understand or know their inputs or representations.

The assumption that LLMs can only echo statistical regularities is important to qualify. So-called 'underclaiming' (downplaying what LLMs do and learn), resonates more broadly in the academic sphere, which could hinder studying how LLMs work in detail (Bowman, 2022b), and exploring whether they are useful for studying questions about human language usage. The assumption likely stems from the idea that probabilistic modelling of language may successfully simulate or approximate linguistic facts (i.e. generate coherent language), but does this via such a different route that it cannot provide any further insight into human language (Norvig, 2012; Piantadosi, 2023).

Although it may seem on the surface that LLMs are just language simulation machines, this overlooks all the linguistic complexity that is stored in the weights that are updated during training, with word prediction providing a powerful supervision signal. Beyond the potential impact of 'underclaiming' in academic debates (Bowman, 2022a), this assumption could reinforce simplistic views of what LLMs are and why they are useful in society at large.

## 2.2 LLMs can master formal aspects of language, but not its function

The distinction between formal and functional language competence we draw on here stems from Mahowald et al. (2023): formal language competence concerns employing information about linguistic rules and patterns in producing coherent output, whereas functional language competence draws on further cognitive capacities, such as formal reasoning, intentional reasoning, and situation modelling. The authors paraphrase this difference as the difference between being good at *language* and being good at *thought*; in their view, LLMs master language but not thought. They motivate this distinction with the finding that the two competences recruit independent brain circuits, and discuss persons with aphasia as a concrete example: they can have limited formal linguistic competence, yet still be able to compose music, solve logic puzzles, reason about other persons' mental states, thus, leverage thought independently of language. Whether one buys into its neural grounding or not, the distinction between formal and formal language competence as such is a useful one to make, and we see similar oppositions in Bender and Koller (2020), where the distinction is made between LLMs' mastery of linguistic form as opposed to extra-linguistic meaning, and in Bisk et al. (2020); Browning and LeCun (2022); Floridi (2023).

### 2.2.1 Disentangling language and thought

Here we do not claim that LLMs have thought that can be meaningfully separated from language, as we are agnostic on this matter, but we question some of the methods currently used to disclose thought.

First of all, whereas persons with aphasia can be tested on their capacity to employ thought in a way that is clearly independent of language (e.g. composing music), for LLMs this is not possible. For example, for the common sense reasoning and intentional reasoning (a.k.a. Theory of

Mind/ToM) humans do, two vital functional capacities, benchmarks inevitably rely on presenting a particular (social) situation using linguistic prompts (e.g. Collins et al., 2022; Creswell et al., 2022; Sap et al., 2022; Binz and Schulz, 2023; Borji, 2023; Kosinski, 2023; Ullman, 2023; van Duijn et al., 2023). Implicitly or explicitly, such works draw on the assumption that in LLMs, there must be a distinction between thought as internal symbolic system representing abstract relations, and language as a mapping between these representations and their outward expression in text. Although this is not unreasonable to think, given that LLMs have many emergent capacities for which they were not explicitly trained (Section 2.1), currently we do not know how much formal linguistic information LLMs leverage when performing such tasks. LLM output could, for example, be the result of specific semantic or syntactic relations with the input, while it seems unlikely that a human would approach such tasks in the same way. Thus, in assessing thought in LLMs, language and thought are confounded.

This issue has an analogy in testing thought in children. When for instance testing ToM, confounding factors are always present, as the myriad tests of ToM that exist and the different modalities they solicit (vision, speech, text) illustrate (Quesque and Rossetti, 2020). General language and memory abilities of children are typically controlled with additional tasks (Milligan et al., 2007); few tests exist that rely on language alone (Beaudoin et al., 2020). Still, many seemingly superficial aspects shape performance on such tests, such as how questions are phrased (Siegal and Beattie, 1991; Beaudoin et al., 2020). The influence of superficial linguistic artefacts of tests can be controlled to some extent when conducting ToM tests with LLMs, for example, preventing memorization by rewriting tests such that they are not in the training data (e.g. Shapira et al., 2023; Kosinski, 2023; van Duijn et al., 2023), but this is only the beginning of disentangling language and thought in LLM output.

Moreover, disentangling language and thought is difficult, because for many cognitive test we have an idea of what the test operationalises, but not when they are used in the context of LLMs. For the ToM context, one hallmark test is the 'unexpected contents' test (Perner et al., 1987), where a Smarties box with unexpected contents (e.g. a pencil) is shown to a child. The child is asked what a friend,

unfamiliar with the box, would think its contents are, thereby asking it to manage two conflicting beliefs: the false belief imputed to the friend, and its own belief about the box' contents. This conflict management, as instantiation of ToM ability, is arguably what the test operationalises, and something humans know from subjective experience, which makes it easier to understand what was measured when such tests are used on humans. Yet, this is far less clear when using such tests on LLMs.

### 2.2.2 Thought is a continuum

Here we assume, for the sake of argument, that we can separate thought from language ability in LLMs with cognitive tests (which we questioned in Section 2.2.1). We further reflect on how various tests are currently being used to deny or affirm thought in LLMs. Again, we do not argue that LLMs have or lack thought here, but rather that we should suspend conclusions, based on the issues raised below.

Quite some cognitive tests currently employed with LLMs make assumptions about thought that need qualification. Many are designed so that a LLM can only fail or succeed on them (e.g. (part of) tests employed by Sap et al., 2022; Borji, 2023; Bubeck et al., 2023; Kosinski, 2023; Shapira et al., 2023; Ullman, 2023). Yet, thought capacity is better understood as a *continuum* (Beaudoin et al., 2020; Sahlgren and Carlsson, 2021). That is, thought capacity is unequally distributed in humans; people who excel in logic, may still be horrible composers, or struggle to recognise other persons' intentional states.

In addition, we are typically much more lenient towards failure in exercising thought. In daily contexts, humans are generally susceptible to a host of misplaced heuristics and formal errors (Haselton et al., 2015; Dasgupta et al., 2022), but we generally do not conclude from this that humans do not have unique thought capacity. Sahlgren and Carlsson (2021) make a similar claim for language understanding: different language users are good at different things, at different times, in different situations. Thus, it is unsurprising that disagreement exists among NLP-scholars about proper operationalisations of various tasks in Natural Language Understanding, such as Natural Language Inference (Subramonian et al., 2023), a disagreement that can also be anticipated for other cognitive tests.

To make our evaluations of thought in LLMs as compelling as possible, we can employ more sophisticated measures, and aim for more nuance in the interpretations of results, before we can deny (or affirm) thought in LLMs. Instead of focusing merely on (average) success or failure, knowing that a LLM has, for example, 51% confidence in a wrong answer is already more informative than just knowing the LLM erred. Fortunately, work on more nuanced evaluations of thought in LLMs is emerging (e.g. Collins et al., 2022; Binz and Schulz, 2023). Alternatively, we could evaluate the model's intermediate reasoning steps in solving a complex reasoning task, besides only the answer, when employing Chain-of-Thought prompting (Wei et al., 2022). Evidence is emerging that in the context of ToM tests, more sophisticated prompting improves performance (Moghaddam and Honey, 2023).

From a methodological perspective, testing thought in LLMs and humans differs a lot because the entities at issue differ a lot. Yet, we can improve the comparability of testing. A single output of a LLM on a single test item likely does not yield a good estimate of its capacities; in testing humans, we typically ask multiple humans to do the same item. Thus, we could for example initialize LLMs with slightly higher temperature values multiple times on the same test item, to get a fuller view on what LLMs can, and to obtain a larger sample of responses on which statistical tests are possible. Although we know that low temperatures settings make models deterministic, not much evaluation of slightly higher temperature settings in relation to performance has been done, with the exception of Moghaddam and Honey (2023). In addition, we know from work on knowledge extraction in LLMs, that paraphrasing a particular input improves model performance in retrieving knowledge and relations (Jiang et al., 2020). Paraphrasing test items is not only a way to increase model performance, but could also provide a way to increase our confidence in our estimates of thought capacity as indicated by LLM performance on multiple paraphrased items.

Lastly, from a more general perspective, failure of a LLM on a cognitive test does not imply that the system does not have the mappings required to do the test; the tests could also be less suited to retrieve them (Bommasani et al., 2021).

### 2.3 Language acquisition in LLMs cannot inform human language acquisition

Bisk et al. (2020) argue that, since children cannot acquire a language by merely listening to the

radio, it is likewise wrong to expect that LLMs can acquire language by purely ingesting text from the internet; similar claims are offered in Bender and Koller (2020); Chomsky (2023). This point relies on the presumed poverty of 'extralinguistic' information in the train data of LLMs. In language acquisition, children draw not only on linguistic input but also on sense perception (e.g. seeing and touching the world), motor experience (e.g. moving objects), and interaction with caretakers (e.g. feedback). Also, children receive far less language input compared to LLMs (Warstadt and Bowman, 2022). Thus, if language acquisition in children and LLMs is so different from the start, it seems language acquisition in LLMs cannot inform language acquisition in humans.

### 2.3.1 LLMs as useful distributional models

LLMs and children evidently differ a lot. Yet, as Sahlgren and Carlsson (2021) formulate, LLMs are theoretically and practically our current 'best bet' for machines to acquire language understanding, given the empirical work documenting LLM proficiency in many language tasks (see e.g. Bommasani et al., 2021; Wei et al., 2023). Like any scientific model they are wrong in some respects, but they seem our current best *distributional* models to study *specific* aspects of language acquisition.

For example, Chang and Bergen (2022) use BERT and GPT-2 as distributional agents that exclusively learn from word co-occurrence statistics. They employ a.o. word frequency, lexical class and word length, as known predictors of word acquisition in children, and predict word acquisition in LLMs and children to gauge the extent to which these known effects in children can be accounted for by statistical learning mechanisms. Chang and Bergen (2022) show that language acquisition in LLMs and children differs in key respects (LLMs are more frequency-driven), but are also similar (learning in both takes longer for words embedded in longer utterances). As the authors note, distributional models can be used in similar fashion to explore the extent to which acquisition of semantics or syntax in children can be accounted for by statistical learning.

Cevoli et al. (2023) provide another example by unravelling lexical ambiguity with BERT. The authors show that psychological theories positing complex mechanisms for representing ambiguity, are not necessary to explain how such representations are acquired, since they can be decoded from distributional information in text. This illustrates another key role LLMs can play in language acquisition: as Warstadt and Bowman (2022) note, LLMs in ablation studies can provide 'proof of concept' whether target linguistic knowledge (e.g. verb-subject agreement in triply embedded clauses) is learnable in an ablated environment (e.g. without triply embedded clauses in the training data). Such studies are helpful in identifying sufficient conditions for obtaining specific linguistic knowledge in language acquisition. Thus, the examples mentioned above echo the broader point made by various scholars that we should see LLMs as distributional learners that show what linguistic phenomena are *in principle* learnable from statistical information in text (Contreras Kallens et al., 2023; Evanson et al., 2023; Wilcox et al., 2023).

Warstadt and Bowman (2022) note that in language acquisition contexts, LLMs need to be less advantaged to humans in one key aspect: the amount of training data. That is, they need to be made more ecologically valid. The latter is indeed important, and fortunately, work is emerging which shows that it is possible to train LLMs with more realistic amounts of data, that at the same time perform equally well in predicting human neural and behavioural patterns as models trained with large datasets (e.g. Hosseini et al., 2022; Wilcox et al., 2023). Yet, it is equally remarkable that LLMs have become so successful, despite being very *disadvantaged* as well (no multimodal input, no feedback in learning, no sensorimotor input). We argue that it is not obvious how we should weigh such disadvantages and advantages in LLMs' language learning. LLMs make wrong assumptions about language acquisition in key respects, but all scientific models do this (Box, 1979; Baayen, 2008), while this does not render such models useless: they can provide a lower bound on what linguistic phenomena are learnable in principle from distributional information.

### 2.3.2 How poor is training data?

Here we discuss the assumption mentioned in Section 2.3 that data used to train LLMs lacks extralinguistic information required in language acquisition, by considering what extralinguistic information LLMs learn to represent during training. Since humans use language to do a variety of things (Sahlgren and Carlsson, 2021), such as providing explanations, describing all sorts of objects and processes, entertaining and convincing others, it is

natural to assume that LLMs are able to recover some of the knowledge about e.g. properties of objects in the world, communicative intents, and users' mental states. Recent work shows that LLMs are able to represent conceptual schemes for worlds (e.g. for direction) they have never observed (Patel and Pavlick, 2022), thus it seems that LLMs have a sufficiently rich conceptual structure to decode at least some of the extralinguistic information present in text, as a surrogate grounding. Similarly, Abdou et al. (2021) show that internal representations of LLMs show a topology of colours that correspond to human perceptual topology. In addition, evidence emerges that LLMs are able to represent communicative intents behind texts (Andreas, 2022), and the ways LLMs represent semantic features of various object concepts aligns with humans (Hansen and Hebart, 2022). Moreover, studies in which LLMs are trained and tested on synthetic tasks, provide an even stricter scenario for testing whether LLMs are able to decode emergent properties from simple input. LLMs trained on simple input such as lists of player moves in a board game, prove able to recover emergent properties such as game rules, valid future moves, and board states (Li et al., 2022). For additional examples of extralinguistic grounding, see Bowman (2023).

## 3  A pragmatic philosophy of LLMs

This section sketches a more general, pragmatist philosophical context for LLMs. Although LLMs are prominent in academia and society, philosophical reflection is lagging behind. This is lamentable, given that LLMs and the way they are deployed raise pressing philosophical questions. Here we develop a pragmatist view on LLMs with the following claims that we will motivated with reference to philosophical pragmatism:

**1.** All three key points from the debate about capacities of LLMs discussed above ultimately revolve around the issue of 'real' understanding and intentionality, but fail to address what that means;

**2.** Once we try to explain what 'real' understanding and intentionality are, we find that these (and mental states more generally) are not accessible in others we interact with, irrespective of whether they are humans or other kinds of systems;

**3.** Attributing mental states to others has foremost pragmatic value, in that they help us to abstract underlying complexity away, predict behaviour, and obtain goals in the world;

**4.** Given the increasing prominence of LLMs, interacting with them in terms of mental state attribution will likely become more common, yet lacks a comprehensive theory;

**5.** This practice is fully explainable from a pragmatist perspective, although in different communities, such as the scientific community, different pragmatist values may play a role, that makes this practice less acceptable for this community.

### 3.1  Invoking 'real' understanding

Various scholars have claimed that LLMs are incapable of 'really' understanding language and using intentionality like humans (e.g. Bender and Koller, 2020; Bishop, 2021; Browning and LeCun, 2022; Floridi, 2023; Mahowald et al., 2023). Indeed, it seems that the critiques of LLMs as autocomplete systems that do not know how language functions in the world, or as language learners that cannot learn by drawing on such functions, implicitly invoke this claim. Still, in such critical works it is seldom made explicit what 'real' understanding or intentionality amounts to. These works often revolve around John Searle's 'Chinese Room' thought experiment (Searle, 1980). We illustrate this with the following quote from Bender and Koller (2020):

> "This means we must be extra careful in devising evaluations for machine understanding, as Searle (1980) elaborates with his Chinese Room experiment: he develops the metaphor of a "system" in which a person who does not speak Chinese answers Chinese questions by consulting a library of Chinese books according to predefined rules. From the outside, the system seems like it "understands" Chinese, although in reality no actual understanding happens anywhere inside the system." (p. 5188)

The argument presented in the quote is that, for any system, being able to deliver the expected output on a range of inputs is insufficient for having 'actual understanding', where it is important to note that the perspective of anyone interacting with the system is 'from the outside'. The point of this thought experiment is that, although the idea of 'real' understanding is implied, it is not explained, which makes the argument incomplete.

## 3.2 Explaining 'real' understanding

The Chinese room argument appeals to a situation where we *would* grant that the system understands Chinese: if the human in the system understands Chinese. That is, this human would need to have a set of *mental states* involving knowledge, beliefs, and intentions, such that in producing output, the human does not draw on predefined rules, but rather on its knowledge of Chinese, beliefs about the desired output, and further communicative intent. This would constitute an example of what is meant by a human having 'real' understanding. Nonetheless, this explanation cannot save the thought experiment as presented above, since it makes no difference for anyone interacting with the system if we would replace the rule-abiding human with the human as full mental agent, since from the outside, there's only the system's behaviour to observe, which does not change.

This distinction between what is observable, e.g. behaviour, and what is inaccessible or unobservable, e.g. mental states, is a distinction known in the philosophy of science (see e.g. Van Fraassen, 1980; Churchland, 1985; Fodor, 1987), but it is also at work in the empirical domain. For example, as Rabinowitz et al. (2018) note with reference to Dennett (1991), we make mental models of others' internal states that are *inaccessible* from the outside, and that make 'little to no reference' to the underlying mechanisms of the agent that produces the observed behaviour. Our point here is that we are *always* confined to observable behaviours of other agents, regardless of whether they are humans or machines. Whenever we claim that 'real' understanding and intentionality is lacking in some other agent, we make a claim about states that are in principle inaccessible from the outside.

## 3.3 Pragmatic value

We are nevertheless fully entitled to make 'mental models' of other humans, that is, attribute to them believes, desires, and intentions, because this is useful in everyday interaction: it has clear *pragmatic value*. This point is perhaps best known in the form worked out by Daniel Dennett as the 'intentional stance': by attributing mental states to other humans, we abstract away from their underlying biophysical complexity, while still having a ground for anticipating future behaviour (Dennett, 1989). If we see a person running towards a bus stop, attributing the desire to catch the bus makes the behaviour intelligible and allows us to predict further behaviour, e.g. waving to the bus driver. Similarly, attributing the set of mental states that constitutes 'real' language understanding to other humans, makes their behaviour intelligible, and smooths our social interactions. This pragmatic perspective is closely related to the idea that mental states are key concepts for humans that have a strong *social* justification (Rorty, 2009); they help a community to achieve its goals in the world, and that is all the justification we need to use them. Exactly because attributing mental states to others has such clear pragmatic value, we have sufficient reason to take them seriously. From this perspective, it is counterproductive to adopt a behaviourist (i.e. denying their importance or existence) or essentialist (i.e. accepting them only if there is evidence that they are 'real') attitude towards mental states.

## 3.4 Pragmatic value and LLMs

We can make a similar claim for LLMs, even though humans and LLMs differ. With regard to observable behaviour, humans can deploy more subtle and multimodal observable behaviours compared to LLMs, like tone of voice, facial expressions, gestures, even unconsciously. So the observable behaviour that underlies our mental models of humans is arguably much richer, which gives us more details to work with when attributing mental states to others. At the same time, we should acknowledge that the way we interact with LLMs is strikingly different from the way we interacted with artificially intelligent systems before. Their language output is grammatically correct, fluent, and critically, increasingly well adapted to context, user, and input. This is starting to challenge assumptions about what it is to be human and what it is to be a machine, and what it is to communicate as a human with an intelligent system that communicates in many ways like a human would do (Guzman and Lewis, 2020). The increasing sophistication of interaction has led to humans viewing such systems as distinct, social entities, and as a consequence, humans are triggered even more to attribute mental states to such systems (see e.g. Guzman and Lewis, 2020; Stuart and Kneer, 2021).

In the context of LLMs, attributing mental states to LLMs has often been addressed as oversensitive anthropomorphisation, with our mental models being illusions 'in the eye of the beholder' (Bender et al., 2021). Such critiques overlook that making

mental models of intelligent systems can have clear pragmatic value, in that they abstract away from the underlying complexities of LLMs, and at the same time help us to predict and explain their behaviour, and achieve goals in the world. Obviously there are complex systems for which making mental models make less sense, e.g. for a Mars rover, where our goal of landing it on Mars is better served by physical models. On the other hand, our interaction with LLMs as complex systems, as we show with examples below, is often best served by attributing mental states to them 'as if' they were socially intelligent in the way we think other humans are.

Our mental model about what an LLM 'knows' or 'wants', can allow us, among other things, to communicate our requests succinctly ('Do you *know* how to do X' in prompting), explain errors ('The system *confuses* X for Y'), formulate a next step in interaction ('The system now *expects* input X'), or gauge reliability of output ('How strong is your *belief* that X?'). And this need not apply only to our own interaction with LLMs, but is also relevant for explaining LLM behaviour to other humans. LLMs optimized for dialogue, (e.g. Chat-GPT, PaLM2-chat), increasingly enable this form of interaction that involves mental state language.

This development should not surprise us, given that language is a tool that has evolved for communicating and manipulating mental states to achieve goals in the world (Clark, 1996; Tomasello, 2003), for example resolving conflict and working together. Similarly, in child development, language competence and the ability to reason about mental states strongly overlap (for an overview see Milligan et al., 2007). Furthermore, scholars argue that children in learning word meanings (for example for verbs of perception like 'to look') do not just learn abstract sign-object mappings (that interlocutor X literally perceives object Y), but foremost their pragmatic effects, which is for children typically directing an interlocutor's attention to various concrete objects (Enfield, 2023; San Roque and Schieffelin, 2019), and such forms of joint attention are a precursor to ToM (Tomasello et al., 1995). In a similar vein, current research that focuses on ways to have LLMs 'reflect' on their 'confidence' in their assertions, or on uncertainty in their input, can be understood in terms of the pragmatic value this has for LLM users, that in the normal world also deal with uncertainty in information (Zhou et al., 2023b; Kadavath et al., 2022).

Given the increasing prominence of LLMs in society, we can expect that making mental models of LLMs will become more common.[2] We can already see some examples where LLMs are specifically used to impersonate individuals, for example, helpdesk service agents (Brynjolfsson et al., 2023), influencers (Lorenz, 2023), deceased beloved ones (Pearcy, 2023), virtual friends (Marr, 2023), and personal assistants (Chen, 2023). These may strike one as rather worrying examples, but such developments could have pragmatic value for humans in that LLMs can give them a sense of relationship or consolation. This is *not* to say that we advocate such deployment of LLMs, as they have many unaddressed ethical implications, but rather that there are conditions imaginable, in which mental state attribution to LLMs is explainable, justified, and has pragmatic value. Here we rather want to stress that the larger role LLMs (in whatever future form) will likely play in society, demands a theory of our interactions with them that does not simplify our behaviour to 'anthromorphisation'.

### 3.5 Pragmatic value and science

We want to emphasise that pragmatic values can be different in different communities, since they may have different goals in the world. In a scientific community that attempts to describe/explain LLMs and their purported cognitive capacities in more detail than is typically required in daily life, researchers may balk at attributing mental states to LLMs. Yet, they should not do so because LLMs do not have any 'real' understanding and intentionality, as we saw that this claim misses the point. Mental states are not intended as literal accounts of the underlying complexity of humans or machines, and the subjective experience associated with 'real' understanding is not something we can access from the outside, and therefore deny outright in other entities.

A better reason, grounded in the pragmatist perspective we offer here, seems to be that mental models made in everyday interaction may allow us to explain and predict behaviour, but lack other pragmatic values critical in the scientific community. If mental states are to play a role in a scientific description or explanation of LLMs, then they must, for example, *also* cohere with other currently accepted theories/models; offer an elegant or simple

---

[2]Note that our account is not intended to be normative, in that we are not claiming that humans should make mental models of LLMs

explanation, have a large scope, etc. (Van Fraassen, 1980). Such values are pragmatic, because they do not primarily depend on the relation a theory has with the observable world; there is, for example, no reason to think that the world must be elegant or simple because our theories are, or that phenomena in the world cohere because our theories cohere.

The upshot is that attributing mental states to LLMs may not cohere well with empirical work on mental states in other fields that map them to patterns of neuronal activity in the brain, for which neurons in LLMs currently constitute at best only a loose analogy. Or it may not cohere with more theoretical work that holds that the possibility of mental states in machines entails a category mistake (as introduced by Ryle (1950)), as mental states are properties of beings such as humans, which fundamentally differ from machines. By considering pragmatic values at play in different communities, we are able to explain why, on a general level, attributing mental models to LLMs can be explainable and justifiable, but at the same time could be less acceptable in the scientific community that has different pragmatic values.

## 4 Discussion

Although in Section 3 we discuss LLM capacities and mental states mostly at a fundamental level, our arguments are also relevant for engineers working on concrete systems that employ LLMs. Such systems will always require reflection on understanding and mental states in humans and machines, which our pragmatic outlook can inform. Our arguments are agnostic about the explicit taxonomies and frameworks of the mental, which engineers may develop and employ in such systems, as it is the system's behaviour that the pragmatist is typically most interested in, and it can be realised in various ways. In the design of LLMs, no such taxonomies or frameworks exist (Kosinski, 2023; Trott et al., 2023), but it is possible that systems that do have them manifest equally complex behaviour. In a similar vein, we can imagine training scenarios that include visual (or other multi-modal) input as a proxy for grounding denotations of words in the world, which would also make LLM behaviour more sophisticated, as disambiguating input is arguably simpler with an additional information channel. Evidence is emerging that enriching LLMs with vision modules as surrogate grounding allows such models to learn new words more efficiently

(Ma et al., 2023).

A related point is that, although a pragmatic account of mental states in humans abstracts away from their complex underlying biophysical correlates, pragmatism does not entail that there is no point in trying to disclose such correlates scientifically, with the aim of opening the black box. A biophysical account of mental states may have pragmatic values for a community of scientists (see Section 3.5), and also broader pragmatic value for society in that it can help us to, for example, treat dysfunctional mental states better. This biophysical account resides at a different level of explanation, and does not necessarily conflict with pragmatic accounts of the mental in general. In the case of LLMs, the pragmatist has similarly no principal issues with trying to find out what patterns of (artificial) neuronal activation are correlated with mental state content in LLM input and output.

## 5 Conclusion

The goal of this paper was to provide further reflection on LLMs in two ways. First, we scrutinised three key points surfacing in recurring critiques on LLMs, and found that on empirical and theoretical grounds, these points need more nuance. Our conclusions are that LLMs are more than exploiters of statistical patterns; that we need better measures for evaluating thought competence in LLMs before we can draw conclusions; and that LLMs have a role to play in language acquisition, as our current best distributional models.

Second, we provided a philosophical context for LLMs from a pragmatist perspective. An unresolved question underlying various critiques of LLMs, is whether they have something like 'real' language understanding and intentionality. We argued that whether we attribute unobservable mental states to other entities, including the set that would constitute 'real' language understanding and intentionality, depends on how much pragmatic value this has to us, not on whether mental states are actual properties of the entities at issue.

LLMs (in whatever future form) will become more prominent in the years to come. We hope to have contributed to a better understanding of what LLMs can(not) do, as well as to a philosophically informed understanding of our interaction with LLMs that is more than a story of mere anthropomorphisation.

## Limitations

In this paper, we addressed LLMs as the set of large Transformer-based neural networks that are trained with cloze tasks, using large text datasets. Still, there is some variation in this set, as LLMs can have different sizes, different architectures, training datasets, methods for further fine-tuning, and so on. Up to this point, it is typically the case that larger LLMs trained with more data obtain the best performance on a variety of tasks, which also makes that such larger LLMs are overrepresented in evaluations of general LLM capacities. OpenAI's flagship models like GPT-3 and ChatGPT are LLMs that frequently recur in tests (although the work of e.g. Shapira et al. (2023) is an exception).

In addition, new models are appearing at a fast pace, such as LLaMA (Touvron et al., 2023), Falcon (Penedo et al., 2023), and PaLM2 (Anil et al., 2023). It remains to be seen how these new models fare on various tests, such as those for cognition, but they are fairly similar regarding their neural network architecture, training data, and training objective. At the same time, signs are emerging that OpenAI's flagship models may be slowly deteriorating with respect to their performance on writing code and doing basic math (Chen et al., 2023).

All these developments together challenge the idea that there is something like 'the' LLM, which is a simplification we made in this paper that is not doing complete justice to the large zoo of LLMs that currently exists. In addition, the continuing updates they are undergoing to make them derail less quickly, safer, less bias-driven, more efficient, and so on, also imply that they are a moving target in many discussions. These fast developments may also limit the import of the arguments into the more distant future, as it is hard to foresee for example developments in different neural architectures and training regimes.

## Acknowledgements

This research was not possible without collaboration with dr. Max van Duijn's research project 'A Telling Story' (with project number VI.Veni.191C.051), which is financed by the Dutch Research Council (NWO). We thank Li Kloostra for helpful comments on earlier versions of this paper. Lastly, the authors thank three anonymous reviewers for their constructive feedback.

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
