# OpenReview forum: "Large Language Models: The Need for Nuance in Current Debates and a Pragmatic Perspective on Understanding"
_EMNLP/2023/Conference — EMNLP 2023 Main_

### Official Review · Reviewer_rDe7 · 2023-08-01

**Soundness:** 3

**Excitement:**

4: Strong: This paper deepens the understanding of some phenomenon or lowers the barriers to an existing research direction.

**Paper Topic And Main Contributions:**

This paper is a position paper (with some aspects of a survey paper). It presents two main arguments. First it argues that more nuance is needed in response to claims that LLMs simply regurgitate statistical patterns. Secondly, it offers a new perspective on the definition of “understanding” and why attributing mental states to LLMs could be useful in terms of pragmatic value. To form these arguments, this paper surveys a multitude of papers ranging from the analysis of LLMs to philosophy of science papers.

**Reasons To Accept:**

This paper makes some convincing arguments. It convincingly argues that the claim that “all LLMs can do is next word prediction” is too simplistic and that underclaiming LLMs capabilities is not useful in academic debates. The paper also brings up an interesting point by saying that when we claim that humans have mental models, we generally also only do so from an outsider perspective. Lastly, the emphasis on more sophisticated LLM evaluation is also very important and the point that thought is a continuum is also very interesting.

**Reasons To Reject:**

A couple of the arguments made in the paper could be made more convincing: The section about language acquisition in LLMs vs. humans (Section 2.3) is confusing and it is unclear what argument is being made about language acquisition. It is also unclear what the message is in Section 2.2.1: the paragraphs seem to contradict themselves.
Additionally, many of the citations (especially in the introduction) are bertology papers and it is unclear how their findings can be applied to more recently published LLMs.

**Reproducibility:**

N/A: Doesn't apply, since the paper does not include empirical results.

**Reviewer Confidence:**

4: Quite sure. I tried to check the important points carefully. It's unlikely, though conceivable, that I missed something that should affect my ratings.

---

> ### Author Rebuttal · Authors · 2023-08-28
>
> **First of all, thank you very much for your time. We are grateful for your positive remarks and the thoughtful and constructive comments. Please find our responses below: suggested changes to the final draft are marked with bullets.**
>
> ## Points to improve
> ---
> ### Regarding Section 2.3 / what LLMs can contribute to language acquisition research:
>
> Research shows that the unparalleled capacity of LLMs to produce coherent text relies at least in part on their capacity to learn complex linguistic features during training (lines 287-388). While it is obvious that there are numerous differences between LLMs and human learners, we argue that their case is still informative for studying specific aspects of language acquisition, such as the question to what degree lexical understanding and semantic ambiguity in children can be theoretically accounted for by statistical learning mechanisms (lines 392-413 and 414-431). LLMs are wrong in key respects (like all models), still they are very advanced distributional learners that can help us find out what linguistic phenomena can *in principle* be accounted for by information extracted from word contexts.
>
> ### Section 2.2.1 / disentangling language and thought:
> The confusion (“the paragraphs seem to contradict themselves”) could follow from our phrasing at the beginning of this section (215-219). We do not want to argue for or against thought in LLMs in this section, but argue that the way we usually try to separate language from thought capacity in humans, is not straightforwardly applicable for LLMs. Lines 215-219 serve as clarification (but is perhaps not clear enough) that we question the *methods* for testing thought in LLMs, but do not intend to use our points as arguments for or against thought capacity (as separate from language) in LLMs.
>
> - We will clarify our aim of this section in lines 215-219.
>
> ### Regarding potential outdatedness of BERT references:
> Given the fast pace of the field right now, it is a good point that this paper requires recalibration as more LLMs come to the fore (e.g. LLaMA2, Falcon-Instruct). We have carefully checked our arguments and references for relevance to recent developments at the time of writing the paper (April-June 2023), and will include an additional round of updates in the final draft. Still, research about the specific linguistic and cognitive capacities of recent models has yet to emerge. In the discussion mentioned in the introduction (lines 44-59), various reserchers voice (theoretical) arguments concerning BERT (e.g. Bender et al. (2021)), but also more recent models like GPT-3, LaMDA (e.g. Bender et al. (2021), Marcus (2022)) and Chat-GPT (e.g. Piantadosi (2023) and Shanahan (2022)).
>
> - We will include an outlook on recent models in the discussion section in the final draft.
>
> ---
> - Emily M. Bender, Timnit Gebru, Angelina McMillan- Major, and Shmargaret Shmitchell. (2021). On the Dangers of Stochastic Parrots: Can Language Models Be Too Big? In Proceedings of the 2021 ACM Conference on Fairness, Accountability, and Transparency, pages 610–623.
> - Gary F. Marcus. (2022). Nonsense on Stilts. Online resource. https://garymarcus.substack.com/p/nonsense-on-stilts Accessed on: 2023-01-30.
> - Murray Shanahan. (2022). Talking About Large Language Models. arXiv preprint arXiv:2212.03551.
> - Steven T. Piantadosi. (2023). Modern language models refute Chomsky’s approach to language. Online resource. https://lingbuzz.net/lingbuzz/007180. Accessed on: 2023-04-15.
> ---
> **We hope that this addresses your points.**

---

### Official Review · Reviewer_6wK9 · 2023-08-04

**Soundness:** 3

**Excitement:**

4: Strong: This paper deepens the understanding of some phenomenon or lowers the barriers to an existing research direction.

**Paper Topic And Main Contributions:**

This work is a position piece arguing for how one should proceed to understand whether neural models understand. In particular, the paper first discusses three main critiques of the capacity of large language models: i) language models as parrots, ii) language models as learning formal and not functional aspects of language, and iii) language models as uninformative w.r.t. language acquisition. The main contribution of this work is a sustained discussion of a particular world view with which to interpret models.

**Questions For The Authors:**

Additional Comments:

A. The paper states “[m]ental states are not intended as literal accounts of the underlying complexity of humans”. This is given no supporting citation, and I do not think this is a universally agreed upon point. I would imagine many theorists actually attribute mental states to humans.


**Reasons To Accept:**

The paper is thought-provoking and articulates an interesting philosophy for understanding neural models. I expect the paper to engender discussion in the field (at least the part of the field interested in how to conceptualize models). Further it is well-written and bridges the gap between NLP and philosophy in ways that are rare in the field right now.

**Reasons To Reject:**

While I enjoyed this paper, there are three things that the paper should consider (assuming that I have correctly understood the paper. If not, I will update my comments).

First, the aim of the paper is a bit unclear. Namely, whether the paper is outlining why people may be led to attribute mental states to LLMs and an articulation of when they should. I believe my confusion stems from statements like “we reflect on the circumstance under which it would make sense for humans to similarly attribute mental to LLMs” and “we ask [when] it is justifiable for humans to make mental models of LLMs”. I read this as either stating conditions that motivate people to make these claims or stating conditions when people ought to make such claims. While I accept the work as possibly accounting for why people make such claims, the work lacks discussion of whether people ought to. Similar issues concern the “pragmatic” aims of treating language models as having mental states with queries (on lines 663 and 667) like “Do you know how to do X” and “How strong is your belief that X”. Under the first interpretation (i.e., mental states are attributed), theses queries seem reasonable. People treat the model as if it has these capacities, so they naturally query it for these states (as they do with other humans). Under the second interpretation (i.e., models ought to be deemed as having mental states), more work should be done to justify why these queries are good. For example, do current models actually have the capacity to respond to queries like this in useful ways? More generally, it actually may be harmful for people to associate mental states to LLMs, as this may lead them to be prone to misinformation, for example. The paper would benefit from engaging with the societal conditions that language models are embedded in.

Second, one argument (as I understand it) in the paper lacks some justification. I read part of the work as asserting something like i) LLMs are intelligent systems, ii) humans best understand intelligent systems by ascribing to them mental states, and iii) therefore LLMs should be deemed as having mental states. The question the field faces, in some part, is whether LLMs are actually intelligent systems, so this first point depends heavily on how one understands (limited) existing empirical evidence. If we read these points as an account for the reasons laypersons assign mental states to LLMs, then as above, some justification is needed for why this actually helps people understand LLMs. Cars and planetary movements are complex, but I don’t think the best account of those is to assign them mental states. Presumably, other factors contribute to when we should do this (e.g., things being “social entities” as on line 647, though again LLMs aren’t necessarily social entities). The work, then, would be to justify why LLMs should be taken as having those abilities. I see this as also dove tailing with point 3 on page 6 -- “attributing mental states…help[s] us to abstract underlying complexity away”. However, we don’t understand all complex things as mental, so why treat LLMs this way?

Third, section 2.2.2 seems to me to be taken as given when the points of it are at issue in the ongoing debates of the field. I think my confusion stems for brining two sense of thought: i) thought as a human capacity for certain types of mental states, and ii) instances of thinking in a colloquial sense (e.g., good at solving riddles). It is stated on lines 297-298, that “thought capacity is unequally distributed in humans”. However, all humans possess thought of type (i) and I take this claim to be of type (ii). Why should differences in things people are good at thinking about be a justification for thinking LLMs have thought?


**Reproducibility:**

N/A: Doesn't apply, since the paper does not include empirical results.

**Reviewer Confidence:**

4: Quite sure. I tried to check the important points carefully. It's unlikely, though conceivable, that I missed something that should affect my ratings.

---

> ### Author Rebuttal · Authors · 2023-08-28
>
> **First of all, thank you very much for your time. We are grateful for your positive remarks and the thoughtful and constructive comments. Please find our responses below: suggested changes to the final draft are marked with bullets.**
>
> ## Reasons to Reject
> ---
> ### 1
> We intend our account not to be normative, but aim to explain why humans can attribute mental states to LLMs, and why it can be rational and productive to do so in certain contexts. This is close to Daniel Dennett’s ‘intentional stance’, as explained in section 3.3. We observe that current LLMs, especially those that are instruction-tuned (e.g. GPT-4, PaLM2), prompt users to engage with them as they would with other humans, incl. queries that pertain to mental states. We do not argue that users *ought* to see such systems as having beliefs and desires, although this behaviour *is* justifiable: it may enable efficient communication or perhaps even a sense of relationship (lines 651-700), thus, have pragmatic value. Although we mention example applications of LLMs (lines 685-700) in society, in which interaction with LLMs using mental state language seems natural, and mention that there are critical ethical concerns, we feel an in-depth discussion of such topics is beyond the scope of the current paper.
>
> - We will review and disambiguate section 3.4 to clarify that our approach is not intended to be normative.
>
> ### 2
> The pragmatist focuses primarily on a system’s (complex) behaviour, and on whether our attribution of mental states helps us achieve our goals (e.g. efficient communication, or perhaps a sense of relationship as mentioned in lines 685-700). This is not because a system might be intelligent, but because interaction with it is best served by assuming that it is. While we argue that this is the case for LLMs, it is not for planets: our interaction with their behaviour is better served by physical models, given our goals such as landing a Mars rover, thus in this case physical models would have more pragmatic value.
>
> - This point will be made more explicit in the final draft, to avoid confusion.
>
> ### 3
> In section 2.2.2., the points we raise about human thought are not primarily intended to claim that LLMs actually have or do not have thought (line 315), but rather to call for some reflection about how testing thought in LLMs is currently done.
>
> If we understand you correctly, you see (i) as the idea that all people have mental states, and (ii) as a form of (commonsense) reasoning. The pragmatist agrees with (i), as it has a strong social justification (lines 611-616) to think that others have mental states. The point regarding (ii) is that the robustness of human reasoning varies with context (lines 300-307), and capacities for subtypes of thought (e.g. social reasoning) may differ among people, yet we are typically more nuanced in drawing conclusions on thought in humans; we e.g. may critically reflect on our tests first, attempt to test skills via other modalities, and so on. We call for the same caution in testing LLMs, but do not aim to assert or deny thought in LLMs with these points.
>
> - We agree that our final draft will benefit from making this point clearer in section 2.2.2.
>
> ## Questions
> ---
> ### A
> One example of AI researchers who employ the same pragmatic idea of mental states as we do, are Rabinowitz et al. (2018). From their introduction:
>
> *"What does it actually mean to “understand” another agent? As humans, we face this challenge every day, as we engage with other humans whose latent characteristics, latent states, and computational processes are almost entirely inaccessible [...] A salient feature of these “understandings” of other agents is that they make little to no reference to the agents’ true underlying structure. A prominent argument from cognitive psychology is that our social reasoning instead relies on *high-level models* of other agents."* [Our italics]
>
> The idea of high-level models stems from Gopnik & Wellman (1992) (quoted by Rabinowitz and colleagues), scholars from developmental psychology, who make a related point about how children develop models of other humans' mental states as *theories* intended to explain and predict behaviour, rather than literal accounts. Regarding language understanding specifically, pragmatism as stance within the philosophy of language, in which language understanding or intentionality is primarily assessed through observable behaviour, is discussed in the review by Bommasani et al. (2022). Note that for a pragmatist, attributing mental states to others is perfectly justifiable if it has pragmatic value (e.g. allows us to explain and predict behaviour), which will be often the case.
>
> - Bommasani et al. (2021). On the Opportunities and Risks of Foundational Models. arXiv preprint, arXiv:2108.07258.
> - Gopnik, A. and Wellman, H. M. (1992). Why the child’s theory of mind really is a theory. Mind & Language, 7(1-2): 145–171.
> - Rabinowitz, N., Perbet, F., Song, F., Zhang, C., Eslami, S.M.A. &amp; Botvinick, M.. (2018). Machine Theory of Mind. Proceedings of the 35th International Conference on Machine Learning, in Proceedings of Machine Learning Research. 80:4218-4227
>
> ---
> **We hope that this addresses your points.**

---

### Official Review · Reviewer_QBMH · 2023-08-05

**Typos Grammar Style And Presentation Improvements:** N/A
**Soundness:** 4

**Excitement:**

4: Strong: This paper deepens the understanding of some phenomenon or lowers the barriers to an existing research direction.

**Missing References:**

**General Ideas:**

It Takes Two to Tango: Towards Theory of AI's Mind. Arjun Chandrasekaran et al.

**Evidence that LMs learn more than the next word:**

[1] Emergent World Representations: Exploring a Sequence Model Trained on a Synthetic Task. Kenneth Li et al. ICLR. 2023

[2] Do Language Models Have Beliefs? Methods for Detecting, Updating, and Visualizing Model Beliefs. Peter Hase et al. 2023. EACL.

**Computational inquiry of LMs as human language learners:**

[1] Using Computational Models to Test Syntactic Learnability. Ethan Gotlieb Wilcox et al. 2023. Linguistic Inquiry.

[2] Language Acquisition: Do Children and Language Models Follow Similar Learning Stages? Linnea Evanson et al. 2023. ACL.

[3] World-to-Words: Grounded Open Vocabulary Acquisition through Fast Mapping in Vision-Language Models. Ziqiao Ma et al. 2023. ACL.


**Paper Topic And Main Contributions:**

Upon the recent debate on the understanding ability of large language models (LLMs), this position paper presents a critical analysis of three prominent criticisms levied against them. These criticisms assert that LLMs are limited in their capacity to grasp concepts beyond distributional semantics, lack functional language competence, and do not aid in human language acquisition. The authors emphasize the need for nuance in the discussion of these issues. In the latter section, the authors offer a philosophical perspective grounded in pragmatic considerations, which serves to justify situations where attributing mental states to LLMs becomes relevant and holds pragmatic significance. By delving into these discussions, the paper seeks to contribute to the community by dispelling unfounded allegations against LLMs and shifting the focus of debates from the ontological nature of understanding to the practical value it entails.

**Questions For The Authors:**

Question A: Does the position paper implies that the selection and definition of mental states for LLMs hold little significance as long as pragmatic value exists?

Question B: Does the position paper implies that researchers can only form theories about how LLM blackbox works, and confirming theories about LLM blackbox workings become extremely challenging, akin to forming theories for human biological processes?

Question C: Can this perspective be extended to future foundation models, such as large vision-language models, that could also be interpreted as agents with a language channel? Why?


**Reasons To Accept:**

Strength 1: This position paper is very fluent and beautifully-written, and presents a delightful read. It masterfully addresses timely issues around LLMs, providing a well-rounded representation of the current state of the field, while also giving due recognition to the main counter-arguments.

Strength 2: The authors offer a philosophical perspective centered in pragmatic values, which potentially helps to dispel unwarranted accusations directed at LLMs. Moreover, this perspective aims to redirect the focus of debates to the pragmatic value it brings to the human users, rather than definition of “real understanding”.


**Reasons To Reject:**

Weakness 1: Lack of connection to concrete action items to build NLP applications with LLMs. The definition of mental states in this work focuses on the latent nature, rather than a concrete taxonomy or a theoretic abstraction framework. While helping researchers to shape a scientific philosophical perspective of language understanding in LLMs, the connection to concrete, grounded applications is relatively weak. The audience would largely be academic researchers with a focus on AI as a science rather than practitioners and researchers who are interested in AI as engineering.

Weakness 2: The pragmatic account of LLMs offers a perspective to simplify and abstract the actual underlying blackbox, and understand their behaviors. On the other hand, it introduces two additional concerns: (1) the selection and definition of mental states for LLMs hold little significance as long as pragmatic value exists, and (2) researchers can only form theories about how LLM blackbox works, and confirm theories about LLM blackbox workings become extremely challenging, akin to forming theories for human biological processes.

Weakness 3: The definition of mental states in this work seems to overwhelmingly favor intentionality. As theory of mind (ToM) is essentially the ability to ascribe mental states, I believe this paper can benefit from a slightly extended discussion on ToM and explainability.


**Reproducibility:**

N/A: Doesn't apply, since the paper does not include empirical results.

**Reviewer Confidence:**

4: Quite sure. I tried to check the important points carefully. It's unlikely, though conceivable, that I missed something that should affect my ratings.

---

> ### Author Rebuttal · Authors · 2023-08-28
>
> **First of all, thank you very much for your time. We are grateful for your positive remarks and the thoughtful and constructive comments. Please find our responses below: suggested changes to the final draft are marked with bullets.**
>
>
> ## Weaknesses
> ---
> ### 1
> Our paper is indeed focused on how we can approach LLM capacities and mental states at a fundamental level. Still, we believe that the pragmatic approach is also relevant for e.g. engineers and users. Interactive systems that show complex behaviour will always ask reflection on the relation between human and machine cognition. Users can use our arguments to explain why e.g. attributing mental states to such systems is defendable. And, because a pragmatic account is mainly focused on the practical value of a system’s complex behaviour, it is liberal on the specific taxonomies/frameworks of the mental employed in realizing behaviour. An engineer can make sense of system behaviour with pragmatist arguments, *even* when no such explicit taxonomy/framework in creating the system is involved (which seems largely the case for LLMs).
>
> - We will explain in the discussion section our conviction that discussion at the fundamental level, will ultimately benefit theorists and practitioners.
>
> ### 2
> Pragmatic accounts of mental states are in no way intended as the only attempt to ‘open the black box’ of LLMs (or humans). Accounts of mental states can operate at different levels of analysis and explanation, similar to how neurobiology may identify different correlates for specific mental states compared to social psychology, yet these need not exclude each other (although their relation is likely complex). Regarding (1), it is not the case that ‘anything goes’ in a pragmatic account, as different explanations have different pragmatic values depending on in which context they are used. A scientific context may require an explanation to meet additional pragmatic concerns, compared to an everyday social context (section 3.5).
>
> - We will make the different levels of analysis explicit in the discussion, possibly with reference to David Marr’s (1982) account of levels.
>
> Marr, D. (1982), Vision: A Computational Approach, San Francisco, Freeman & Co.
>
> ### 3
> We agree that a broader discussion of ToM could be an interesting extension for this paper; as we could discuss more extensively e.g. i) why people attribute mental states to others at all (what are its benefits in the context of evolution, lines 670-678), and ii) what further benefits ToM has (e.g. advanced reasoning about mental states such as embedded mindstates, in addition to benefits mentioned at lines 651-684).
>
> - If this matches with your idea of explainability, we will include these points in section 3.4.
>
> ## Questions
> ---
> ### A
> This question is mostly addressed in the discussion of Weakness 2: for the pragmatist there is no objection towards further study of correlates of mental states in LLMs (or humans), although this is indeed a challenging task. Nor would the pragmatist object to creating different types of systems (symbolic, RL-based, language models, etc.) that attempt to conceptualize or model intentionality. They operate at different levels of analysis, which will be made clearer in the final draft of the paper.
>
> ### B
> Pragmatism does not object against forming and testing theories about how LLMs work, yet it offers a workable approach to go by while the extremely challenging task of mapping out the mechanisms of LLMs’ ability for e.g. modelling intentions of users is ongoing.
>
> ### C
> Very interesting; we think the short answer is yes, for the same reasons set out in 3.1 and 3.2 of the paper. Visual information could additionally help for example to disambiguate information by ‘grounding’ (albeit as a proxy) distinctions common in the verbal realm with an additional information channel, which likely boosts pragmatic value of the system’s output.
>
> - We will add reflection on the applicability of our approach to future models in our discussion.
>
> ---
> **We hope that this addresses your points. Thank you also for the pointers to additional relevant literature; we will carefully check the papers and consider them for the final draft.**

---

### Meta-Review · Area_Chair_2gSx · 2023-09-15

**Recommendation:** 5

**Metareview:**

Reviewers gave scores of 4,3,3 (soundness) and 4,4,4 (excitement).

Strengths and weaknesses including the following were prominent:

Strengths:

- well written (R1) and thought-provoking (R2, R3)

Weaknesses

- lack of connection to concrete implications for NLP applications using LLMs (R1)
- questions were raised about the philosophical underpinnings and justification of some claims (R1, R2, R3)

---

### Decision · Program_Chairs · 2023-10-07

**Decision:**

Accept-Main

**Comment:**

Reviewers gave scores of 4,3,3 (soundness) and 4,4,4 (excitement).

Strengths and weaknesses including the following were prominent:

Strengths:

- well written (R1) and thought-provoking (R2, R3)

Weaknesses

- lack of connection to concrete implications for NLP applications using LLMs (R1)
- questions were raised about the philosophical underpinnings and justification of some claims (R1, R2, R3)